# An exploratory study on excess weight gain: Experiences of Postmenopausal Women in Ghana

Isaac Mensah Bonsu[1]*, Hellen Myezwa[1], Corlia Brandt[1], Adedayo Tunde Ajidahun[1], Monday Omoniyi Moses[2], Benjamin Asamoah[2]

1 Department of Physiotherapy, School of Therapeutic Sciences, University of The Witwatersrand, Johannesburg, South Africa, 2 Faculty of Allied Health Sciences, Department of Physiotherapy and Sports Science, College of Health Sciences, Kwame Nkrumah University of Science and Technology, Kumasi, Ghana

* piceabc@yahoo.com

## Abstract

### Introduction

Excess weight gain is a problem with a significant impact on health and quality of life as well as the consequent economic burden on human populations. While society advocates preference for excess weight gain, limited evidence exists concerning postmenopausal women's experiences in Ghana.

### Aim

The current study explored the experiences regarding excess weight (overweight and obesity) gain among Ghanaian postmenopausal women in Bono East (Techiman) region.

### Methods

This is a qualitative exploratory descriptive study where anthropometric measurements [body mass index (BMI), waist-to-height ratio (WHtR) and waist-to-hip ratio (WHR)] were determined and participants who were identified with excess weight (obesity and overweight) were recruited to participate in a focus group discussion (FGD). Four focus group discussions were held and 24 postmenopausal women (>45 years) in Techiman took part. Discussions were audio-recorded and transcribed for thematic analysis.

### Results

From the qualitative analysis of the FGD, three major themes emerged from the study namely, (i) perception of body weight, (ii) measures to reduce body weight, and (iii) support to adhere to the weight management program. Sub-themes revealed that weight gained was viewed as the presence of diseases and admired by society as a culturally accepted body image. Diet-related changes, a combination of diet modification and physical activity, and weight management supplements emerged as measures to reduce excess weight.

**Data Availability Statement:** All relevant data are within the manuscript.

**Funding:** The author(s) received no specific funding for this work.

**Competing interests:** The authors have declared that no competing interest exist

Support to adhere to the weight program included health education, access, affordable exercise facilities, and social support.

## Conclusions

Sociocultural norms influence the image perception and acceptance of weight gain among postmenopausal women in Ghana, but there is an acknowledgment of the perceived negative health implications of such excess weight. Measures for weight reduction and support to adhere to the weight reduction programme require attention in Ghana.

## Introduction

Excess weight (overweight and obesity) refers to a build-up of adipose tissue (body fat) that causes deficiency in the well-being, physical health, and emotional health of an individual [1]. Globally, the prevalence and trend of excess weight gain are of major economic concern, as behaviour alterations have led to this unhealthy body weight [2]. The global prevalence of excess weight among post-menopausal women is about 10% [3]. In Ghana, the prevalence of excess weight gain is between 10.7% and 14.1% and women are particularly affected within the West African region [3].

Excess weight gain impacts the quality of life of an individual [4]. Studies revealed that individuals who are obese are adversely impacted and experience direct and indirect criticism and judgement from others [5–10]. Consequent negative feelings cause depression and anxiety among obese individuals [4]. Additionally, negative personal beliefs about inferiority are commonly associated with obese individuals [11–13]. Weight gain is a health concern for women especially those aged 55–65 years [14]. This is understandable as the occurrence of obesity is increasing globally and nearly 300 million women are obese [15]. The prevalence of weight gain is common during the menopausal transition [16]. Independent of women's early body size or race/ethnicity, women gain about 0.7 kg per year on average during the fifth and sixth decades of lifespan [16]. Additionally, the sharp increase in excess weight is due to numerous factors and is only in part attributable to the adoption of a western lifestyle (physical inactivity and excessive consumption of energy-dense food) observed in the last 20 years [17]. Other studies have demonstrated that the disruption of the circadian cycle, skipping breakfast, regular eating occurrence, unbalanced meals and consumption of large food size, could predict obesity [18,19].

Weight gain among postmenopausal women is associated with the appearance of excess central body fat, mostly visceral fat [18,19]. These associated changes are due to an estrogen deficiency level after menopause, ageing and factors that influence a healthy lifestyle [14]. Abdominal obesity has adverse metabolic concerns, such as hypertension and cardiovascular disease [20]. Given that weight gain is a major risk factor for coronary heart disease, infarction, and stroke and causes death in postmenopausal women [19,21], it is important to explore postmenopausal women's experiences of their weight gain.

Several studies have established that one's perception of body size differs with ethnicity, and culture [22–24] and can influence weight gain [25–28]. In the developed world, the thin body dimension is associated with attractiveness, and smartness and represents perfect body image [29]. However, studies conducted in African countries have shown that excess weight gain is associated with cultural standards and accepted appearance that signifies attractiveness, social success, beauty and wealth [30–33]. For instance, Holdsworth et al., [34] studied healthy and

desirable body size perception among urban Senegalese women. Out of 301 women sampled, the majority believed being overweight is appropriate and symbolizes socially desirable body size.

The public health interventions to address excess weight gain in Ghana have focused on diet-related changes and the adoption of physical activity [6]. Little is known about the experiences and factors influencing behaviour around weight gain of postmenopausal women and how these can be incorporated into management approaches in the Ghanaian population. An individual's body weight explains the long-term energy balance between the calories consumed from food and the energy expended [35]. The rise in urbanization and westernization in West Africa are two frequently mentioned factors for the change in the energy balance [36]. Theoretically, as societies get more urbanized and westernized, people engage in less physical exercise and have access to a wider variety of high-calorie fast foods and beverages [37]. Understanding the experiences of postmenopausal women can assist policymakers and public health services to develop evidence-based interventions to manage weight gain in the menopausal transition.

### Aim

The current study explored the experiences regarding excess weight (overweight and obesity) gain among Ghanaian postmenopausal women in Bono East (Techiman) region.

## Materials and methods

### Study design

An exploratory, descriptive qualitative research design using focus group discussion (FGD) was used to explore the experiences of postmenopausal women with excess weight gain in the Bono East (Techiman) region of Ghana.

### Study setting

The study was conducted in Techiman, the capital of the Bono-East region of Ghana. Techiman is one of the leading market towns in the country and one of the two main cities and settlements of the Bono-East region of Ghana [38]. There is diversity in cultural and social structure in the town due to the numerous economic activities engaged by the settlers [39]. These features made the town suitable for this study. A preliminary study was conducted to determine the lifestyle, sociocultural factors and prevalence of excess weight gain among postmenopausal women in Techiman. The study established a higher prevalence of excess weight gain among postmenopausal women using WHtR (91.8%), and WHR (91.0%) and is due to cultural beliefs, and a sedentary lifestyle [unpublished data]. Therefore, developing culturally appropriate interventions for postmenopausal women with weight gain will be helpful.

### Study population and sample

A purposive sampling method was used to recruit participants who satisfied the inclusion criteria. The participants were recruited through public invitation (such as information centres, announcements at churches, Techiman main markets and mosques) where their anthropometric measurements [body mass index (BMI), waist-to-height ratio (WHtR) and waist-to-hip ratio (WHR)] were measured, and those who have excess weight (obesity or overweight) were purposively selected to be part of FGDs after endorsing the informed consent form. The sample frame comprised post-menopausal women in Techiman municipality in the Bono East Region of Ghana, aged 45 years and above and with a BMI $> 25kg/m^2$. The sample size

included twenty-four participants (n = 24) assembled into four focus groups with an average of six members per group.

## Inclusion and exclusion criteria

Our study included postmenopausal Ghanaian women over 45 who were not on any weight-loss therapy and had no physical or mental disabilities. Women who are not Ghanaians and had not resided in Ghana for the previous three years before the study were not eligible.

## Data collection tool and procedure

Moderator's guide (S1 File) for the focus group discussion was used as the data collection tool. The authors developed the guiding questions based on the study objective and available litera-ture. During the interview information on their thoughts regarding their current weight, things they do to manage their weight and type of assistance to adhere to weight management programmes were gathered. The validity of the information to be gathered was determined by an expert in women's health (CB). Additional probes were used to gain a further understand-ing of the participants' experiences of weight gain.

A research assistant who spoke and wrote Twi (the local language) was consulted to trans-late the interview guide. Pre-interview demographic data (age, religion, marital status, ethnic origin and educational background) were collected using a self-administered questionnaire for each session of the focus group before the discussion. The FGDs were facilitated by a trained moderator who was aided by a research assistant. The average duration for each FGD was about 1 hour. The FGDs were all scheduled on a date other than the recruitment date at their convenient time and place. Dieticians and exercise scientists were consulted to offer nutri-tional and exercise counselling services for participants who needed support after sharing their experiences. Almost all the participants used these services. The discussion was audio-taped with the participants' consent. The first author obtained the recording and listened to the dis-cussion immediately after each focus group session. Inconsistent comments were probed for understanding. The audio recordings from the discussion were transcribed by an independent translator fluent in both the Twi and English languages. Transcribed focus group discussions were provided to two randomly selected participants from each group to read through their responses for validation. The transcripts were independently read repeatedly by the first author (IMB) and the fifth author (BA) to make sense of the experiences of postmenopausal women with excess weight gain. Field notes were used for each session to capture the participants' non-verbal cues, apprehensions, and the author's reflections that gave context to the data. The field notes were reviewed, and reports were prepared for each focus group session.

## Data analysis

A thematic analysis process was done in three phases namely pre-analysis, exploration, treat-ment and interpretation to analyse the data [40]. To generate data that directly relate to the experiences of postmenopausal women with excess weight gain, important frequent phrases or statements were extracted from transcripts as codes (S2 File). Related or similar codes were bundled into categories to help organize the data effectively. Formulated meanings were then generated from the significant phrases or statements. Subsequently, themes, sub-themes and categories that emerged were noted based on the multiple statements that conveyed similar meanings. We explored individual themes and attempted to determine possible interrelated-ness between themes.

The first (IMB), third (MOM) and fifth (BA) authors coded all the focus group transcrip-tions whiles the second (CB) and sixth (HM) authors read the transcripts and reviewed the

fragments that resulted from the coding process. During the coding phase, we focused on three domains of interest: body weight, weight loss approaches and adherence to weight management. Each author (IMB, MOM, BA) independently coded one domain from the four transcripts. The codes were presented to the moderator for verification. Eventually, the authors deliberated about the coding and key themes that emerged. Field notes, made by IMB during the discussion, were used to supplement the results. Trustworthiness and credibility in this study were ensured by peer de-briefing and member checking [41]. At the end of each focus group session, member checks were carried out by summarising key responses for participants to validate if they accurately reflected participants' true experiences living with excess body weight. The first author also gathered the responses of each focus group interview to facilitate an exact representation of participants' views and experiences.

## Ethical considerations

Ethical approval for this study was obtained from the Human Research Ethics Committee at the University of the Witwatersrand, South Africa (Ref no. M190467). In Ghana, approval was also obtained from the Committee on Human Research, Publication and Ethics of Kwame Nkrumah University of Science and Technology, Kumasi (Ref No. 596/19). The aim, risks and benefits of the study were explained to the participants during the recruitment stage. Participants were informed that their participation is voluntary and that they reserve the right to withdraw from the study without any prejudice. Those who consented signed the consent forms and were guaranteed their anonymity in future reports and publications. Participants were informed that their contribution will help identify measures to manage excess weight among postmenopausal women.

## Results

### Demographic characteristics of the participants

Twenty-four postmenopausal women consented to participate in the FGDs. Most of the participants were married and from the Akan tribe. The study included postmenopausal women, age range 45 to 74 years, a majority of whom were Christians (54.2%) and had at least a high school certificate (54.2%). The Participant's demographic characteristics and anthropometric data are summarised in Table 1.

### Main findings

From the qualitative analysis of the FGD, three major themes emerged from the study namely, perception of body weight, measures to reduce body weight, and support to adhere to the weight management programme, which is briefly described in Table 2 below.

### Perceptions of current body weight

The theme contextualises the postmenopausal woman's personal experiences of excess body weight (obesity and overweight). Two underlying sub-themes emerged from this main theme which includes (1) admirable/ presentable; and (2) the presence of diseases.

### Sub-theme: Admirable/Presentable

Respondents expressed that their motivation for desiring some amount of body weight was the ability to fit well in their clothes during traditional functions like funerals, wedding ceremonies and others and these indicate a sign of "enjoying good living." The following quotes speak to this sub-theme.

**Table 1. Demographic information of participants.**

| Demographics | |
|---|---|
| **Age in years, Mean (SD)** | 58.00 (7.21) |
| **Age group (years), n = 24(%)** | |
| 45–55 | 10 (41.7%) |
| 56–66 | 11 (45.8%) |
| ≥67 | 3 (12.5%) |
| **Religion, n = 24(%)** | |
| Christian | 13 (54.2%) |
| Muslim | 11 (45.8%) |
| **Marital Status, n = 24(%)** | |
| Married | 14 (58.3%) |
| Widow | 7 (29.3%) |
| Divorced | 3 (12.5%) |
| **Educational background, n = 24(%)** | |
| Primary | 7 (29.2%) |
| High School | 13 (54.2%) |
| Tertiary | 4 (16.7%) |
| **Ethnic origin, n = 24(%)** | |
| Northerner | 8 (33.3%) |
| Akan | 13 (54.2%) |
| Ga | 1 (4.2%) |
| Voltarian | 2 (8.3%) |
| **Anthropometrics** | |
| **Focus group one (n = 24)** | |
| *Mean±SD* | 60.33±4.46 |
| *Mean/BMI* | 29.1 |
| *Mean/WHR* | 0.88 |
| *Mean/WHtR* | 0.59 |
| **Focus group two (n = 24)** | |
| *Mean±SD* | 63.66±7.66 |
| *Mean/BMI* | 32.4 |
| *Mean/WHR* | 0.92 |
| *Mean/WHtR* | 0.61 |
| **Focus group three (n = 24)** | |
| *Mean±SD* | 56.00±4.00 |
| *Mean/BMI* | 31.5 |
| *Mean/WHR* | 0.92 |
| *Mean/WHtR* | 0.60 |
| **Focus group four (n = 24)** | |
| *Mean±SD* | 58.66±6.28 |
| *Mean/BMI* | 30.9 |
| *Mean/WHR* | 0.90 |
| *Mean/WHtR* | 0.63 |

> *I see myself as very active and admirable even though I have gained weight. I feel presentable when I attend functions like funerals, church and wedding ceremonies because I fit well in my cloth. I do almost all my activities myself. Most people admire me because of my weight and the activity I use to do work. I'm seen as being healthy and stronger than some of my age mates.* (FGD 1, participant #3*)*

**Table 2. Themes and sub-themes generated.**

| THEMES | SUB-THEMES |
|---|---|
| **Perceptions of body weight** | • Admirable/Presentable<br>• Presence of diseases |
| **Measures to reduce excess weight** | • Diet-related changes<br>• Combination of diet modification and physical activity<br>• Weight management supplement |
| **Support to adhere to the weight management programme** | • Health Education<br>• Access and affordable exercise facilities<br>• Social support |

*As for me, I take pride in myself and how I look. My weight is not a challenge to me in any way. When you feel good about yourself and who you are, you stand tall and naturally carry yourself with a sense of confidence and self-acceptance that makes you beautiful regardless of your body weight, size, or shape.* (FGD 2, participant #4)

*I am not concerned because I feel good about myself and am active. I am comfortable with my body weight and am full of confidence when I step out to anywhere.* (FGD 3, participant #1)

## Sub-theme: Presence of diseases

The participants in this study anticipate some negative health implications of excess weight among postmenopausal women. This understanding is expressed in the quotes by two respondents who indicated that their excess body weight status was responsible for them having health challenges.

*I'm not as healthy as I used to be. I have health challenges and chronic diseases such as hypertension, waist pain and recently diagnosed diabetes. I have taken pain relief medications for several years but still, the pain comes when the medication gets finished.* (FGD 2, participant #1)

A 56-year-old woman lamented that,

*My weight makes me feel tired easily even if I do fewer demanding activities. I feel heaviness in my body which makes me uncomfortable. I have severe knee and waist pains and I am spending a lot of money on medications.* (FGD 4, participant #4)

## Measures to reduce body weight

This theme explored current weight management practices among postmenopausal women. Three measures emerged as subthemes that informed the postmenopausal weight management strategies. They include (1) diet-related changes; (2) diet modification and physical activity; (3) weight management supplements.

## Sub-theme: Diet-related changes

They stated that they have reduced the intake of starchy food and meat and now enjoy seasonal fruits are grown locally, such as watermelon, orange, and mango, which were previously absent from their diet.

*I have reduced my intake of starchy foods and meat. I am very selective about the kind of food I eat recently. I eat vegetables and fruits often to manage my body weight. I started these about a year ago and have seen some significant changes.* (FGD 3, participant #5)

*I take some fruits such as watermelon, mango, orange, etc. I don't eat after 5 pm. I have a backyard garden, so I have access to fruits. I also don't take salt and fresh meat.* (FGD 4, participant #6)

## Sub-theme: Diet modification and physical activity

Participants highlighted an amalgam of dietary and physical activity modification as a way of weight management. As part of the participants' weight management strategies, they indicated changes in diet and physical activity as an approach. Two participants commented as follows:

*I control the quantity of food I eat now. I now eat heavy food in the morning and take fruits in the evening and engage in physical activities too. I have been going to the farm also and I believe all these are part of physical activities.* (FGD 4, participant #4)

*I eat dinner at 5:00 pm and walk for about 30 minutes every evening around my compound. I also walk to serve my customers, instead of using a car. I participate in outdoor games occasionally.* (FGD 2, participant #2)

*I like fufu (Ghanaian indigenous starchy food) very well and eat it every day, I have only reduced the size of the fufu. I eat some seasoned fruits like watermelon and mangoes. I have not seen any significant reduction in my body weight. I also walk most of the time to church and meetings.* (FGD 4, participant #1)

One participant confessed that, even though the size of her meal has not reduced, she engaged in some form of physical activity such as walking and sweeping and occasionally participate in outdoor games during holidays to manage her excess weight as shown below:

*I have not been able to reduce the quantity of food I take but I take in more of our local greens, fruit and dry fish almost every day, especially on weekends. I also walk up and down to serve my customers most often.* (FGD 3, participant #5)

## Sub-theme: Weight management supplement

Unlike other women who used diet modifications and physical activity as measures to manage body weight, some participants were motivated by weight management supplements, pills and teas which they believed are a fast and best option to achieve their weight management agenda. According to them, they use some of the supplements to prepare tea for breakfast without any sugar and milk as shown in the following narrations:

*I am taking weight management supplements like herbal life as breakfast without sugar and milk. I take these products because I want to reduce my weight quickly. I take this twice a day with a required dosage for about four months now.* (FGD 4, participant #6)

*I drink slim tea in the morning and forever living products, clean 9. I started this about four months ago. I need a quicker way to reduce my weight because my blood pressure is not stable, and I'm told to do something about my weight.* (FGD 1, participant #3)

*I drink a mixture of warm water and honey every morning. A friend introduced this method to me, and I believe it will work for me. I am told I can also take warm water and lemon; I will try that one too.* (FGD 3 participant #3)

*I started with herbal life supplement a year ago, however, I recently bought slim tea to manage my weight because I wanted fast results. A friend introduced a forever-living supplement two weeks ago. These supplements are expensive, so I moved to natural means like a mixture of honey and lemon. I drink boiled lemon every morning before I step out* (FGD 2, participant #4)

## Support to adhere to the weight management programme

The theme describes the various supports to aid weight management programs. In this study, the sub-themes generated under this theme are (1) health education on lifestyle modification, (2) access to available and affordable exercise facilities, and (3) self-control and motivation.

### Sub-theme: Health education

Participants raised a need for diet-related education including timing and type of food to eat. They also mentioned education that highlights the importance and type of exercise as measures of adherence to exercise programs. Some of the women cited they are oblivious to the risk of excess body weight and how to avoid its consequences and thus, health education through the local radio stations and hospital information centers can assist them to have all the necessary information to modify their lifestyle. This is illustrated in the narratives below:

*We need health education in our homes through radio stations and public health systems. These will help us to reduce the causes of some diseases.* (FGD 1, participant #1)

*We will need education from qualified health professionals on how to modify our lifestyles such as eating habits and the dangers of not exercising. These will help our weight management program.* (FGD 2, participant #4)

*Doctors should constantly come to our end and educate us on the dangers attached to excess body weight and ways and means to prevent it.* (FGD 4, participant #2)

In one of the groups, one respondent reported that the media and health workers should provide education on the efficacy of weight management supplements in the market. The FGD participant recounted bad experiences of consuming certain weight management supplements of which she had little knowledge and suggested that education on this topic is much needed.

*We would be very grateful if the information is given on the effective weight management supplement that we should buy. I remember a friend introduced me to a weight management supplement called slim tea and I had a bad experience. I had diarrhea for a whole night and was admitted to the hospital for three days before discharge.* (FGD 3, participant #1)

### Sub-theme: Access and affordable exercise facilities

Participants cited that the availability of affordable exercise facilities encourages them to adhere to weight management programs.

*We need gym facilities in the community that are accessible and not expensive. This will encourage us to exercise.* (FGD 2, participant #5)

*There must be exercise facilities in our community which are cheap with qualified people to assist us* (FGD 3, participant #4*)*

### Sub-theme: Social support

Other participants suggested encouragement from family members and society by offering positive words of support and approval can assist their adherence to the weight management program as described below:

*Inspiration from family members and society can help us adhere to a weight management program.* (FGD 4, participant #5*)*

*Motivational words such as you are looking good, you have reduced in weight, and your health issues are getting better from family especially our partners and sometimes physicians will encourage us to adhere to a weight management program.* (FGD 4, participant #1*)*

## Discussion

This study aimed to explore the experiences of postmenopausal women with excess weight gain. Three themes were identified from our study and included perception of body weight, weight management practices and support to adhere to weight management programmes.

Excess body weight is usually deemed to be an indication of an unhealthy lifestyle, indicative of the onset of some non-communicable illness. In our study excess weight is viewed within the Ghanaian context as admirable and a culturally accepted norm of financial well-being. The cultural dynamics in Ghana tend to celebrate curvaceous and buxom women because it is seen as a sign of well-being [6,42]. In this regard, women in this cultural setting tend to flaunt their weight gain since the cultural ethos value such a large body size [42]. Participants surmised that traditional ceremonies such as funeral and marriage engagements present rare opportunities to showcase their beauty, thus, a plump and curved woman who fits beautifully in her dress is highly praised and admired. The perception motivates lots of postmenopausal women to accept and flaunt the weight gained during this phase of their lives since the culture celebrates them. This may fuel the epidemic of obesity and overweight in women which are risk factors for cardiovascular diseases, joint pain and other lifestyle diseases such as type 2 diabetes [43]. This affirms earlier findings that describe the perception of body weight as socially admirable and presentable [6]. The onset of postmenopausal symptoms is linked to a higher risk of chronic diseases which is a factor of concern when occurring with obesity or excess weight gain. The finding in our study namely, that being overweight or obese, indicates the presence of a disease, is consistent with a study that reviewed exercise guidelines and considerations in postmenopausal women with obesity. They found an association between the transition to menopause and increased heart disease and osteoporosis [43].

It is therefore important to consider the findings in the study which highlighted the current weight management practice such as diet-related changes. In the management of excess weight, participants narrated that they adopted dietary adjustments. This finding could be explained by the fact that participants saw dietary modification as a strategy for weight loss and its implication on health improvement and disease prevention [44], consequently modifying the quantity and type of diet. This finding, thus, has implications for health professionals especially in public health. Dieticians should educate postmenopausal women on how to manage their diet to prevent excess weight gain.

In addition to the measures to reduce body weight, participants also mentioned that they normally adopted a combination of physical activity and alteration of their diet to reduce their

excess weight gain. This finding indicates that regular physical activity such as walking, sweeping, stair climbing, and appropriate nutrition could be the main determinants of a healthy lifestyle in this population [44]. Previous authors established that reducing body mass index, losing weight and visceral adiposity through diet changes and physical activity, have been shown to reduce the risk and incidence of chronic diseases as well as type 2 diabetes more effectively [43,45]. Remarkably, some participants wanted to reduce their weight following the findings of Aryeetey [6] and in most cases, they wanted to reduce their body weight as fast as possible. Our results show that several management techniques such as supplements, diet-related changes and physical activities are being used by postmenopausal women, in their attempt to reduce their excess body weight. Most of the time, the efforts and efficacy of these therapies do not produce successful outcomes [6]. In Ghana, the Ministry of Health developed physical activity and diet guidelines, however, it has failed to recommend the prevention of excess weight gain in post-menopausal women. The evidence suggests the urgent need to develop policies and regulate the weight management industry in the country. The policies will guide the citizens and health care professionals towards evidence-based therapies for weight management [6].

Support to adhere to the weight management program identified included health education, access and affordable exercise facilities and social support. A 12-month study conducted by Acharya et al [46] to describe participants' adherence to multiple components of a standard behavioural treatment program revealed adherence rates as low as 10%. Thus, to increase adherence rates, the program should incorporate an aspect of social support. Social support (whether through family, friends or peers) is important for successful weight management and is a significant determinant in overall health [46] as it serves as a motivational tool for the women to remain involved in the weight reduction program. Our findings correspond with earlier studies which posits that social support is necessary for weight reduction programs and it encourages participants to adhere to weight management programmes [46,47].

Additionally, access to and availability of affordable exercise facilities are essential to supporting physical activity engagement as it decreases some emotional and physical barriers to exercise [48]. The finding of our study has implications for policymakers to create an environment that supports physical activity in the community (e.g., Recreational parks) and incorporates technology to track behaviour change. Again, when it comes to weight programs, the significance of health education cannot be overlooked [49]. Consequently, there is a need for public health departments and community health educators to address the community's specific needs regarding diet and physical activity through educational programs.

## Conclusion

The present study has illuminated the experiences of postmenopausal women with excess weight gain in Ghana. Even though excess body weight is a threat to the health of an individual, it is admired and socially accepted by the society within the Ghanaian cultural context as we have established in the Bono-East (Techiman) region. Considering the measures to manage or reduce excess weight, there is a need for policies and regulations in the weight management industry to guide the public, particularly women. Additionally, incorporating aspects of social support will encourage adherence rates in weight management programs.

## Supporting information

**S1 File. Moderator's guide.**
(DOCX)

**S2 File. Coding tree.**
(DOCX)

## Author Contributions

**Conceptualization:** Isaac Mensah Bonsu, Hellen Myezwa, Corlia Brandt.

**Data curation:** Isaac Mensah Bonsu, Corlia Brandt.

**Formal analysis:** Isaac Mensah Bonsu, Hellen Myezwa, Adedayo Tunde Ajidahun, Benjamin Asamoah.

**Methodology:** Isaac Mensah Bonsu, Benjamin Asamoah.

**Supervision:** Hellen Myezwa, Corlia Brandt, Adedayo Tunde Ajidahun, Monday Omoniyi Moses.

**Validation:** Hellen Myezwa.

**Visualization:** Isaac Mensah Bonsu, Hellen Myezwa.

**Writing – original draft:** Isaac Mensah Bonsu, Hellen Myezwa, Corlia Brandt, Adedayo Tunde Ajidahun.

**Writing – review & editing:** Isaac Mensah Bonsu, Hellen Myezwa, Corlia Brandt, Adedayo Tunde Ajidahun, Monday Omoniyi Moses, Benjamin Asamoah.

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
