## [Decision Letter · Decision Letter 0]

1 Aug 2022

PONE-D-22-09660An exploratory study on excess weight gain: Experiences of Postmenopausal Women in GhanaPLOS ONE

Dear Dr. Mensah Bonsu,

Thank you for submitting your manuscript to PLOS ONE. After careful consideration, we feel that it has merit but does not fully meet PLOS ONE’s publication criteria as it currently stands. Therefore, we invite you to submit a revised version of the manuscript that addresses the points raised during the review process.

Please note that we have only been able to secure a single reviewer to assess your manuscript. We are issuing a decision on your manuscript at this point to prevent further delays in the evaluation of your manuscript. Please be aware that the editor who handles your revised manuscript might find it necessary to invite additional reviewers to assess this work once the revised manuscript is submitted. However, we will aim to proceed on the basis of this single review if possible. They request some clarifications on the manuscript, could you please revise your article to address their concerns?

We look forward to receiving your revised manuscript.

Kind regards,

Thomas Tischer

Staff Editor

PLOS ONE

https://journals.plos.org/plosone/s/file?id=ba62/PLOSOne_formatting_sample_title_authors_affiliations.pdf".

“No funding”

Reviewers' comments:

Reviewer's Responses to Questions

**Comments to the Author**

1. Is the manuscript technically sound, and do the data support the conclusions?

Reviewer #1: Yes

2. Has the statistical analysis been performed appropriately and rigorously? 

Reviewer #1: N/A

3. Have the authors made all data underlying the findings in their manuscript fully available?

Reviewer #1: Yes

4. Is the manuscript presented in an intelligible fashion and written in standard English?

Reviewer #1: Yes

5. Review Comments to the Author

Reviewer #1: Many thanks for inviting me to review the manuscript titled “An exploratory study on excess weight gain: Experiences of postmenopausal women in Ghana”. Overall, the findings could shade lights on the potential influential components on developing appropriate interventions for managing excess weight gain in the postmenopausal women.

Nonetheless, some parts of the manuscript warrants clarification. I have the following comments for the authors to consider while revising their manuscript:

Abstracts:

• Please clarify if “the society” stated in introduction and result refers to the Ghanaian society rather than society as a whole.

• Suggest to bring forward the three main themes earlier in the result section.

Methods:

• Please clarify what inclusion criteria was used to define postmenopausal women in the study.

• Page 7, moderator’s guide is commonly used for FGD rather than semi-structured interview guide.

• Please clarify whether thematic analysis or content analysis was used. It was stated in page 7 that thematic analysis was performed whilst in abstract and page 8 (data analysis) that thematic content analysis was used. Note that there is subtle difference between thematic analysis and content analysis approaches (see: Vaismoradi et al, 2013, DOI:10.1111/nhs.12048).

• Stated on page 8 that de-briefing and member checking were performed to ensure trustworthiness and credibility of the study. Please provide details on how these processes were conducted.

• Page 8, I don’t understand the statement “The first author also surmised the responses of each focus group interview to facilitate an exact representation of participants’ views and experiences”. Please clarify how this was performed.

• Please provide the moderator’s guide as a supplementary document.

Results:

• Page 9, total participants of the study was 24, if n=24, should it be 100%? Please rectify the frequency and percentage for “Christians” and “high school certificate”.

• Suggest to add participants’ anthropometric data in Table S1.

• Page 11, sub-theme – diet-related changes. Consider to add information on participants’ previous diet in the description, e.g. they reduced the intake of starch and meat, and have now switches their diet to mainly fruits and vegetable.

• Page 12, suggest to provide explanation on “fufu” in bracket, this will help international readers' better understand the context.

Discussions:

• Page 18, stated that “Most of the times, the efforts and efficacy of these therapies do not produce successful outcomes”. Please provide evidence or references to support your claim.

6. PLOS authors have the option to publish the peer review history of their article (what does this mean?). If published, this will include your full peer review and any attached files.

Reviewer #1: **Yes: **Yuet Yen Wong

---

## [Author Response · Author response to Decision Letter 0]

25 Aug 2022

Responses to academic editor and reviewer

Dear Editor,

Please find below the responses to the academic editor and reviewer on the manuscript title: An exploratory study on excess weight gain: Experiences of Postmenopausal Women in Ghana. 

We sincerely thank the Reviewer for the truly helpful comments, which we have read through carefully.

Please find our detailed responses below for each comment.

The academic editor

Question 2

a. Comment: Please clarify the sources of funding (financial or material support) for your study. List the grants or organizations that supported your study, including funding received from your institution

Response: We received Faculty Research Committee Individual Grant from the University of the Witwatersrand, Johannesburg

b. Comment: State what role the funders took in the study. If the funders had no role in your study, please state: “The funders had no role in study design, data collection, and analysis, decision to publish, or preparation of the manuscript.”

Response: The funders had no role in study design, data collection, and analysis, decision to publish, or preparation of the manuscript

c. Comment: If any authors received a salary from any of your funders, please state which authors and which funders.

Response: No author received funds for the study

d. If you did not receive any funding for this study, please state: “The authors received no specific funding for this work.”

Response: The authors received no specific funding for this work

Question 3

a. Comment: In your Data Availability statement, you have not specified where the minimal data set underlying the results described in your manuscript can be found.

Response: This has been revised and found on Page 21

Reviewer Comments to the Author

Abstracts

1. Comment: Please clarify if “the society” stated in the introduction and result refers to the Ghanaian society rather than society as a whole.

Response: This refers to Ghanaian society and most African societies. 

2. Comment: Suggest to bring forward the three main themes earlier in the result section

Response: This has been revised and highlighted in RED on Pages 2 and 3

Methods

1. Comment: Please clarify what inclusion criteria was used to define postmenopausal women in the study.

Response: 

Women who have not had their period for 12 consecutive months and over the age of 45 who were not on any weight-loss therapy and did not have any physical or mental disabilities. 

Comment: Page 7, moderator’s guide is commonly used for FGD rather than semi-structured interview guide.

Response: This has been revised and highlighted in RED on Page 7

Comment: Please clarify whether thematic analysis or content analysis was used. It was stated in page 7 that thematic analysis was performed whilst in abstract and page 8 (data analysis) that thematic content analysis was used.

Response: This has been revised as “thematic analysis” in the abstract and highlighted in RED 

Comment: Stated on page 8 that de-briefing and member checking were performed to ensure the trustworthiness and credibility of the study. Please provide details on how these processes were conducted.

Response: As part of the debriefing process, participants were informed about the intentions of the study. Soon after each focus group session ends, the moderator read the excerpts of the discussion and exchanged insights from the focus group session including field notes taken throughout the discussion by the assistant moderator with the participants. 

For member checking, key responses were summarized for participants to validate if they accurately reflected their true experiences. 

Comment: Page 8, I don’t understand the statement “The first author also surmised the responses of each focus group interview to facilitate an exact representation of participants’ views and experiences”. Please clarify how this was performed.

Response: The first author who is fluent in both Twi and English obtained the recording and listened to the discussion immediately after each focus group session. Inconsistent comments were probed for understanding. Field notes were reviewed and reports were prepared for each focus group session. Transcribed focus group discussions were provided to 2 randomly selected participants from each group to read through their responses for validation. The first author also shared the transcription for verification with other authors who were present at the focus group session.

Comment: Please provide the moderator’s guide as a supplementary document.

Response: Thank you. This has been revised and highlighted in RED 

Results 

Comment: Page 9, total participants of the study was 24, if n=24, should it be 100%? Please rectify the frequency and percentage for “Christians” and “high school certificate”.

Response: Thank you for the comment, however, the percentage was used to get insight into the proportion of participants in the study population who are Christians and have completed high school. 

Comment: Suggest to add participants’ anthropometric data in Table S1.

Response: Thank you. This has been revised and highlighted in RED 

Comment: Page 11, sub-theme – diet-related changes. Consider to add information on participants’ previous diet in the description, e.g. they reduced the intake of starch and meat, and have now switches their diet to mainly fruits and vegetable.

Response: Thank you. This has been revised and highlighted in RED

Comment: Page 12, suggest to provide explanation on “fufu” in bracket, this will help international readers' better understand the context.

Response: Thank you. This has been revised as “Ghanaian indigenous starchy food”. Page 13

Discussions

Comment: Page 18, stated that “Most of the times, the efforts and efficacy of these therapies do not produce successful outcomes”. Please provide evidence or references to support your claim.

Response: Please, reference has been provided and highlighted in RED

---

## [Decision Letter · Decision Letter 1]

26 Oct 2022

PONE-D-22-09660R1An exploratory study on excess weight gain: Experiences of Postmenopausal Women in GhanaPLOS ONE

Dear Dr. Mensah Bonsu,

Thank you for submitting your manuscript to PLOS ONE. After careful consideration, we feel that it has merit but does not fully meet PLOS ONE’s publication criteria as it currently stands. Therefore, we invite you to submit a revised version of the manuscript that addresses the points raised during the review process.

We look forward to receiving your revised manuscript.

Kind regards,

Sandra Boatemaa Kushitor, Ph.D.

Academic Editor

PLOS ONE

Journal Requirements:

Additional Editor Comments (if provided):

Dear Authors,

Please revise your manuscript. I have made detailed comments on the pdf attached. The manuscript should be reviewed by an English editing institution that to reduce the grammatical errors in the document. The authors should also use the COREQ checklist for reporting qualitative studies.

Reviewers' comments:

Reviewer's Responses to Questions

**Comments to the Author**

1. If the authors have adequately addressed your comments raised in a previous round of review and you feel that this manuscript is now acceptable for publication, you may indicate that here to bypass the “Comments to the Author” section, enter your conflict of interest statement in the “Confidential to Editor” section, and submit your "Accept" recommendation.

Reviewer #1: (No Response)

2. Is the manuscript technically sound, and do the data support the conclusions?

Reviewer #1: Yes

3. Has the statistical analysis been performed appropriately and rigorously? 

Reviewer #1: N/A

4. Have the authors made all data underlying the findings in their manuscript fully available?

Reviewer #1: (No Response)

5. Is the manuscript presented in an intelligible fashion and written in standard English?

Reviewer #1: Yes

6. Review Comments to the Author

Reviewer #1: Many thanks for inviting me to review the revised manuscript. Of note, some of the previously raised comments were addressed in the author’s responses; however, changes were not made in the revised manuscript. Please make corrections accordingly for the previously raised items as below:

Abstracts:

• Please clarify if “the society” stated in introduction and result refers to the Ghanaian society rather than society as a whole.

Methods:

• Please clarify what inclusion criteria was used to define postmenopausal women in the study.

• Page 8, I don’t understand the statement “The first author also surmised the responses of each focus group interview to facilitate an exact representation of participants’ views and experiences”. Please clarify how this was performed.

In this round of review, I have several additional minor comments/suggestions for the authors to consider in improving the manuscript.

Introduction:

• Please provide a reference citation for the statement on page 5, paragraph 3, i.e. “The public health interventions to address excess weight gain in Ghana have focused on diet related changes and the adoption of physical activity.”

Methods:

• Please add some detail information on how data analysis was performed. Please include who performed the codings? Any software was used?

Results:

• Please integrate participants’ demographic characteristics and anthropometric data into one single table.

• Please add a brief sentence in the paragraph, e.g. the Participants demographic characteristics and anthropometric data are summarised in Table 1.

• Page 12, line 2…enjoy seasonal fruits are grown locally, please remove “are”.

Discussions:

• Page 17, paragraph 2, “…..a finding supported in previous studies [24]”. The reference citation suggests that there was only one previous study. Please cross-check.

7. PLOS authors have the option to publish the peer review history of their article (what does this mean?). If published, this will include your full peer review and any attached files.

Reviewer #1: **Yes: **Yuet Yen Wong

---

## [Author Response · Author response to Decision Letter 1]

18 Nov 2022

Responses to the academic editor 

Dear Editor,

Please find below the responses to the academic editor on the manuscript title: An exploratory study on excess weight gain: Experiences of Postmenopausal Women in Ghana. 

We sincerely thank the academic editor for the truly helpful comments, which we have read through carefully.

Please find our detailed responses below for each comment.

Abstracts:

Comment: Please clarify if “the society” stated in introduction and result refers to the Ghanaian society rather than society as a whole.

Response: Please, this refers to the Ghanaian society 

Methods 

Comment: Please clarify what inclusion criteria was used to define postmenopausal women in the study.

Response: Please, the study included postmenopausal Ghanaian women over the age of 45 who were not on any weight-loss therapy and did not have any physical or mental disabilities. Found on page 7

Comment: Page 8, I don’t understand the statement “The first author also surmised the responses of each focus group interview to facilitate an exact representation of participants’ views and experiences”. Please clarify how this was performed.

Response: Please, “The first author was an observer in all session. He obtained the recordings and listened to the discussion immediately after each of the focus group session. He probed inconsistent comments for understanding. The first author reviewed field notes and prepares reports for each focus group session. After transcription, he randomly selects two participants from each group to read through their responses for validation. 

Introduction 

Comment: Please provide a reference citation for the statement on page 5, paragraph 3, i.e. “The public health interventions to address excess weight gain in Ghana have focused on diet-related changes and the adoption of physical activity.”

Response: Please, the reference has been provided. Found on page 7 “Aryeetey RN. Perceptions and experiences of overweight among women in the Ga East District, Ghana. Frontiers in nutrition. 2016 Jun 2; 3:13.”

Comment: Please add some detailed information on how data analysis was performed. Please include who performed the coding. Any software was used?

Response: Please, no software was used. Data analysis was manually done. Each author (IMB, MOM, BA) independently coded one domain from the four transcripts. The codes were presented to the moderator for verification. Coding tree attached 

Comment: Please integrate participants’ demographic characteristics and anthropometric data into one single table.

Response: Please, this has been done. 

Comment: Please add a brief sentence in the paragraph, e.g., the Participants demographic characteristics and anthropometric data are summarized in Table 1.

Response: Please, this has been revised 

Comment: Page 12, line 2…enjoy seasonal fruits are grown locally, please remove “are”.

Response: Please, this has been revised 

Discussions

Comment: Page 17, paragraph 2, “…..a finding supported in previous studies [24]”. The reference citation suggests that there was only one previous study. Please cross-check.

Response: Please, this has been revised.

---

## [Editor Report · Decision Letter 2]

24 Nov 2022

An exploratory study on excess weight gain: Experiences of Postmenopausal Women in Ghana

PONE-D-22-09660R2

Dear Mr. Bonsu,

We’re pleased to inform you that your manuscript has been judged scientifically suitable for publication and will be formally accepted for publication once it meets all outstanding technical requirements. Complete all the comments found in the attached file. 

Kind regards,

Sandra Boatemaa Kushitor, Ph.D.

Academic Editor

PLOS ONE

Additional Editor Comments (optional):

Dear Author,

Please make the changes I have suggested in the attached file.
---

## [Editor Report · Acceptance letter]

2 Dec 2022

PONE-D-22-09660R2 

An exploratory study on excess weight gain: Experiences of Postmenopausal Women in Ghana 

Dear Dr. Mensah Bonsu:

I'm pleased to inform you that your manuscript has been deemed suitable for publication in PLOS ONE. Congratulations! Your manuscript is now with our production department. 

Kind regards, 

on behalf of

Dr. Sandra Boatemaa Kushitor 

Academic Editor

PLOS ONE